# Beyond GANs: Transforming without a Target Distribution

## Abstract

While generative neural networks can learn to transform a specific input dataset into a specific target dataset, they require having just such a paired set of input/output datasets. For instance, to fool the discriminator, a generative adversarial network (GAN) exclusively trained to transform images of black-haired *men* to blond-haired *men* would need to change gender-related characteristics as well as hair color when given images of black-haired *women* as input. This is problematic, as often it is possible to obtain *a* pair of (source, target) distributions but then have a second source distribution where the target distribution is unknown. The computational challenge is that generative models are good at generation within the manifold of the data that they are trained on. However, generating new samples outside of the manifold or extrapolating "out-of-sample" is a much harder problem that has been less well studied. To address this, we introduce a technique called *neuron editing* that learns how neurons encode an edit for a particular transformation in a latent space. We use an autoencoder to decompose the variation within the dataset into activations of different neurons and generate transformed data by defining an editing transformation on those neurons. By performing the transformation in a latent trained space, we encode fairly complex and non-linear transformations to the data with much simpler distribution shifts to the neuron's activations. Our technique has the advantage of being generally applicable to a wide variety of data domains, modalities, and applications. We first demonstrate it on image transformations and then move to our two main applications in biology: removal of batch artifacts representing unwanted noise and modeling the effect of drug treatments to predict synergy between drugs.

## 1 Introduction

A common situation arises when we have two datasets and seek to learn a transformation that is a mapping from one (the source) to the other (the target). While much existing work has been done on this, less studied is the case where we want to learn a transformation from this source/target pair of datasets and apply it to a second source dataset for which there is no known target.

If the second source distribution differs from the first source distribution, any transformation naively learned with a neural network on only the first source/target pair will suffer from problems including domain shift (the second source distribution systematically differs from the first) and overfitting (which aspects of the target only exist because it started as the first source distribution, and shouldn't be part of the learned transformation?).

This problem is important to address, as learning transformations is a common task in many different contexts, and often it is infeasible or impossible to obtain known source/target pairing information for every distribution needing to be transformed. For example, many experiments in biology are conducted to study the effect of a treatment on a set of samples, such as tissues from different patients. However, due to their expense and difficulty, clinical trials are often performed on only a small subset of the samples. The challenge is isolating treatment-induced variation from the confounding sample-specific variation.

We propose a neural network-based method for learning a transformation that is general enough to be used across a wide range of data modalities and applications, from image-to-image translation to treatment in the biological setting. Popular neural network architectures like GANs pose the problem

as one of learning to *output* data in the region of the space occupied by the target distribution, no matter where the input data is coming from. To fool the discriminator, the generator's output must end up in the same part of the space as the target distribution. The discriminator does not take into account the input points into the generator in any way.

Instead, we reframe the problem as learning a transformation *towards* the target distribution that is more sensitive to where the input data starts. Thus, we could learn an edit between one source and target pair, and apply it to a second source without needing to assume it has no systematic differences from the first source.

We propose to learn such an edit, which we term *neuron editing*, in the latent space of an autoencoder neural network with non-linear activations. First we train an autoencoder on the entire population of data which we are interested in transforming. This includes both the paired source/target data and the second source data. Neuron editing then involves extracting observed differences between the source/target activation distributions for neurons in this layer and then applying them to the second source data to generate a synthetic second target dataset. Performing the edit node-by-node in this space actually encodes complex multivariate edits in the ambient space, performed on denoised and meaningful features, owing to the fact that these features themselves are complex non-linear combinations of the input features.

Neuron editing is a general technique that could be applied to the latent space of any neural network, even GANs themselves. We focus exclusively on the autoencoder in this work, however, to leverage its denoising ability, robustness to mode dropping, and superior training stability as compared to GANs. We demonstrate that neuron editing can work on a variety of architectures, while offering the advantages of introducing no new hyperparameters to tune and being stable across multiple runs.

While latent space manipulation has been explored in previous work, ours differs in several ways. For example, Radford et al. (2015) represents a transformation between two distributions as a single constant shift in latent space. In addition to assuming the latent transformation is the same for all points in the distribution, Upchurch et al. (2017) also uses an off-the-shelf pre-trained Imagenet classifier network. Our work, on the other hand, does not require a richly supervised pre-trained model; also, we model the shift between two distributions as a complex, non-constant function that learns different shifts for different parts of the space. We compare to this "constant-shift" approach and demonstrate empirically why it is necessary to model the transformation more complexly.

Some neurons are not heavily edited but still influence the output jointly with those neurons that are edited due to their integration in the decoding layers, propagating their effect into the output space. Thus, even a relatively simple transformation in the internal layer of an autoencoder allows for modeling complex transformations in the ambient data space.

This aspect of neuron editing draws close connections with the field of domain adaptation, where the goal is to learn features on one labeled dataset that meaningfully separate points in another, unlabeled dataset (Tzeng et al., 2017). Similarly to that task, we want to learn a transformation from the known source to the known target samples that will also apply to the second source dataset where the target is unknown. Thus, neuron editing represents an extension of domain adaptation, where instead of learning a classifier that can be used on the unlabeled data, we are learning a distribution transformation that can be used on the unlabeled data. Further differences include that while domain adaptation attempts to make features for the unlabeled dataset overlap with those of the labeled dataset, neuron editing transforms the second source dataset without first trying to align it to the first source dataset (Sankaranarayanan et al., 2018). Also, different from many domain adaptation techniques, we do not need any sort of pre-trained classifier to yield an informative feature map for the data, as we learn our autoencoder *de novo* (Long et al., 2017). Given the near exclusive focus of the domain adaptation community on learning classifiers on labeled data and applying it to unlabeled data, we are excited to expand the field to also learning transformations on data with known targets and applying it to data with unknown targets.

We demonstrate that neuron editing extrapolates better than generative models on two important criteria. First, as to the original goal, the predicted change on the second source dataset more closely resembles the predicted change on the original source dataset. Second, the editing process produces more complex variation, since it simply preserves the existing variation in the data rather than needing a generator to learn to create it. We compare to standard GAN approaches, dedicated parametric statistical methods used by computational biologists, and alternative autoencoder frameworks. In

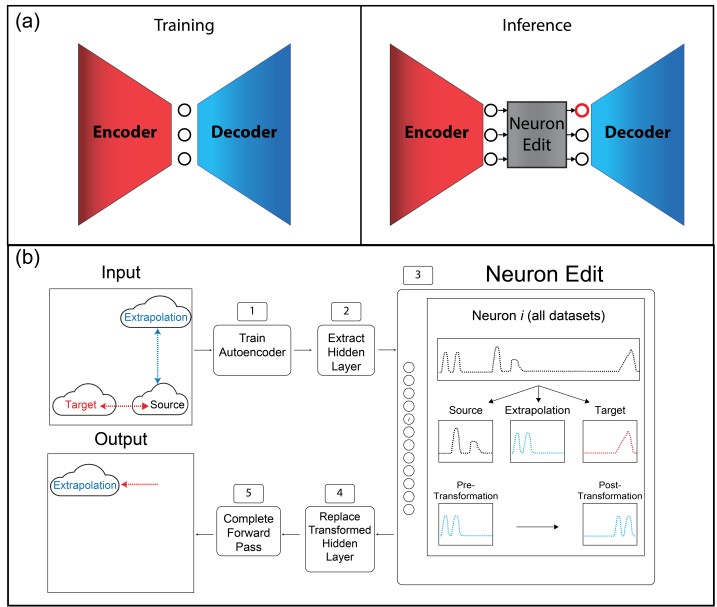

Figure 1: (a) Neuron editing interrupts the standard feedforward process, editing the neurons of a trained encoder/decoder to include the source-to-target variation, and letting the trained decoder cascade the resulting transformation back into the original data space. (b) The neuron editing process. The transformation is learned on the distribution of neuron activations for the source and applied to the distribution of neuron activations for the extrapolation data.

each case, we see that they stumble on one or more of several hurdles: out-of-sample input, desired output that differs from the target of the training data, and data with complex variation.

In the following section, we detail the neuron editing method. Then, we motivate the extrapolation problem by trying to perform natural image domain transfer on the canonical CelebA dataset (Liu et al., 2018). We then move to two biological applications where extrapolation is essential: correcting the artificial variability introduced by measuring instruments (batch effects), and predicting the combined effects of multiple drug treatments (combinatorial drug effects) (Anchang et al., 2018).

## 2 MODEL

Let $S \in \mathbb{R}^{n_S \times d}, T \in \mathbb{R}^{n_T \times d}, X \in \mathbb{R}^{n_X \times d}$ represent $d$-dimensional source, target, and second source distributions with $n_S$, $n_T$, and $n_X$ observations, respectively. We seek a transformation such that: 1. when applied to $S$ it produces a distribution equivalent to $T$ 2. when applied to $T$ it is the identity function and 3. when applied to $X$ it does not necessarily produce $T$ if $S$ is different from $X$. While GANs learn a transformation with the first two properties, they fail at the third property due to the fact that $T$ is the only target data we have for training, and thus the generator only learns to output data like $T$. Therefore, instead of learning such a transformation parameterized by a neural network, we learn a simpler transformation on a *space learned by a neural network* (summarized in Figure 1).

We first train an encoder/decoder pair $E/D$ to map the data into an abstract neuron space decomposed into high-level features such that it can also decode from that space, i.e., the standard autoencoder objective $L$:

$$L(S, T, X) = \text{MSE}\left[(S, T, X), D(E(S, T, X))\right]$$

where MSE is the mean-squared error. The autoencoder is trained on all three data distributions $S$, $T$, and $X$ and thus learns to model their joint manifold. Then, without further training, we separately extract the activations of an $n$-dimensional internal layer of the network for inputs from $S$ and from $T$, denoted by $a_S : S \to \mathbb{R}^n, a_T : T \to \mathbb{R}^n$. We define a piecewise linear transformation, called

$NeuronEdit$, which we apply to these distributions of activations:

$$NeuronEdit(a) = \left( \frac{a - p_j^S}{p_{j+1}^S - p_j^S} \cdot (p_{j+1}^T - p_j^T) \right) + p_j^T \tag{1}$$

where $a \in \mathbb{R}^n$ consists of $n$ activations for a single network input, $p_j^S, p_j^T \in \mathbb{R}^n$ consist of the $j^{th}$ percentiles of activations (i.e., for each of the $n$ neurons) over the distributions of $a_S, a_T$ correspondingly, and all operations are taken pointwise, i.e., independently on each of the $n$ neurons in the layer. Then, we define $NeuronEdit(a_S) : S \to \mathbb{R}^n$ given by $x \mapsto NeuronEdit(a_S(x))$, and equivalently for $a_T$ and any other distribution (or collection) of activations over a set of network inputs. Therefore, the $NeuronEdit$ function operates on distributions, represented via activations over network input samples, and transforms the input activation distribution based on the difference between the source and target distributions (considered via their percentile disctretization).

We note that the $NeuronEdit$ function has the three properties we stated above: 1. $NeuronEdit(a_S) \approx a_T$ (in terms of the represented $n$-dimensional distributions) 2. $NeuronEdit(a_T) = a_T$ 3. $NeuronEdit(a_X) = NeuronEdit(a_S) \implies a_X = a_S$. This last property is crucial since learning to generate distributions like $T$, with a GAN for example, would produce a discriminator who encourages the output to be funneled as close to $T$ as posssible no matter where in the support we start from.

To apply the learned transformation to $X$, we first extract the activations of the internal layer computed by the encoder, $a_X$. Then, we edit the activations with the neuron editing function $\hat{a}_X$. Finally, we cascade the transformations applied to the neuron activations through the decoder without any further training. Thus, the transformed output $\hat{X}$ is obtained by:

$$\hat{X} = D(NeuronEdit(E(X)))$$

We emphasize that at this point, since we do no further training of the encoder and decoder, and since the neuron editing transformation has no weights to learn, there is no further objective term to minimize at this point and the transformation is fully defined.

Crucially, the nomenclature of an *autoencoder* no longer strictly applies. If we allowed the encoder or decoder to train with the transformed neuron activations, the network could learn to undo these transformations and still produce the identity function. However, since we freeze training and apply these transformations exclusively on inference, we turn an autoencoder into a generative model that need not be close to the identity.

Training a GAN in this setting could exclusively utilize the data in $S$ and $T$, since we have no real examples of the output for $X$ to feed to the discriminator. Neuron editing, on the other hand, is able to model the variation intrinsic to $X$ in an unsupervised manner despite not having real post-transformation data for $X$. Since we know *a priori* that $X$ will differ substantially from $S$, this provides significantly more information.

Furthermore, GANs are notoriously tricky to train (Salimans et al., 2016; Gulrajani et al., 2017; Wei et al., 2018). Adversarial discriminators suffer from oscillating optimization dynamics (Li et al., 2017), uninterpretable losses (Barratt & Sharma, 2018; Arjovsky et al., 2017), and most debilitatingly, mode collapse (Srivastava et al., 2017; Kim et al., 2017; Nagarajan & Kolter, 2017). Under mode collapse, significant diversity that should exist in the output of the generator is lost, instead producing synthetic data that is a severely degenerated version of the true target distribution.

Neuron editing avoids all of these traps by learning an unsupervised model of the data space with the easier-to-train autoencoder. The essential step that facilitates generation is the isolation of the variation in the neuron activations that characterizes the difference between source and target distributions.

There is a relationship between neuron editing and the well-known word2vec embeddings in natural language processing (Goldberg & Levy, 2014). There, words are embedded in a latent space where a meaningful transformation such as changing the gender of a word is a constant vector in this space. This vector can be learned on one example, like transforming *man* to *woman*, and then extrapolated to another example, like *king*, to predict the location in the space of *queen*. Neuron editing is an extension in complexity of word2vec's vector arithmetic, because instead of transforming a single point into another single point, it transforms an entire distribution into another distribution.

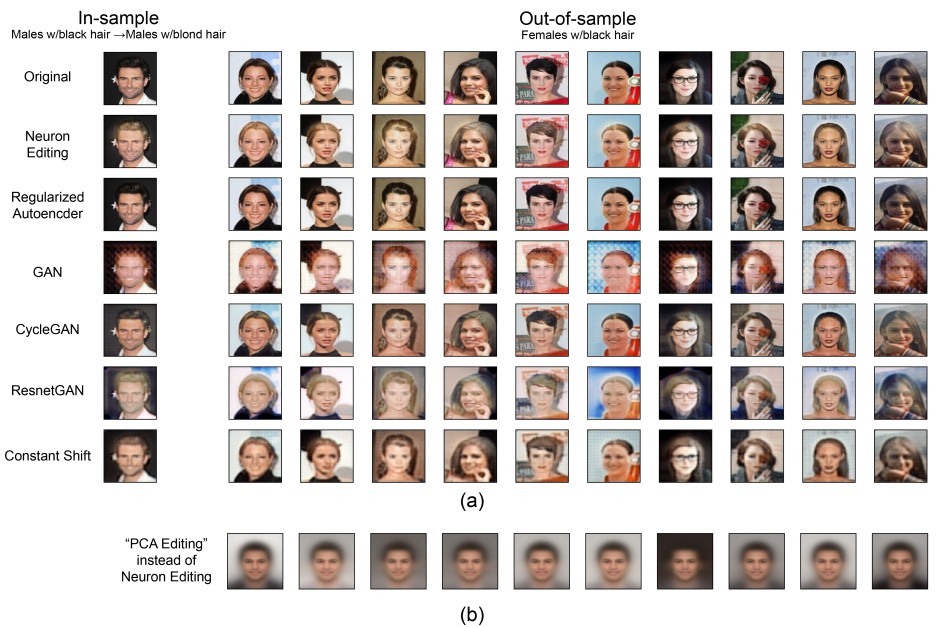

Figure 2: Data from CelebA where the source data consists of males with black hair and the target data consists of males with blond hair. The extrapolation is then applied to females with black hair. (a) A comparison of neuron editing against other models. Only neuron editing successfully applies the blond hair transformation. (b) An illustration that neuron editing must be applied to the neurons of a deep network, as opposed to principle components.

| CelebA | Neuron Editing | GAN | CycleGAN | ResnetGAN | RegAE | Constant Shift |
|--------|---------------|-----|----------|-----------|-------|----------------|
| FID | **121.63 +/- 2.12** | 282.26 +/- 13.32 | 153.03 +/- 6.55 | 184.31 +/- 9.71 | 272.12 +/- 1.10 | 320.97 +/- 1.04 |

Table 1: FID scores on the CelebA extrapolation task.

# 3 EXPERIMENTS

We compare the predictions from neuron editing to those of several generation-based approaches: a traditional GAN, a GAN implemented with residual blocks (ResnetGAN) to show generating residuals is not the same as editing (Szegedy et al., 2017), and a CycleGAN (Zhu et al., 2017). While in other applications, like natural images, GANs have shown an impressive ability to generate plausible individual points, we illustrate that they struggle with these two criteria. We also motivate why neuron editing is performed on inference by comparing against a regularized autoencoder that performs the internal layer transformations during training, but the decoder learns to undo the transformation and reconstruct the input unchanged (Amodio et al., 2018). Lastly, we motivate why the more complex neuron editing transformation is necessary by comparing against a naive "latent vector arithmetic" approach. We find the constant vector between the mean of the source and the mean of the target in the internal layer of our pre-trained autoencoder, and apply this single shift to all neurons in the target (Constant Shift).

For the regularized autoencoder, the regularization penalized differences in the distributions of the source and target in a latent layer using maximal mean discrepancy (Amodio et al., 2018; Dziugaite et al., 2015). The image experiment used convolutional layers with stride-two filters of size four, with $64$-$128$-$256$-$128$-$64$ filters in the layers. All other models used fully connected layers of size $500$-$250$-$50$-$250$-$500$. Leaky ReLU activation was used with $0.2$ leak. Training was done with minibatches of size $100$, with the Adam optimizer (Kingma & Ba, 2014), and learning rate $0.001$.

### 3.1 CelebA Hair Color Transformation

We first consider a motivational experiment on the canonical image dataset of CelebA (Liu et al., 2018). If we want to learn a transformation that turns a given image of a person with black hair to that same person except with blond hair, a natural approach would be to collect two sets of images, one with all black haired people and another with all blond haired people, and teach a generative model to map between them. The problem with this approach is that the learned model may perform worse on input images that differ from those it trained on. This has troubling consequences for the growing concern of socially unbiased neural networks, as we would want model performance to go unchanged for these different populations (Tatman, 2017).

This is illustrated in Figure 2a, where we collect images that have the attribute male and the attribute black hair and try to map to the set of images with the attribute male and the attribute blond hair. Then, after training on this data, we extrapolate and apply the transformation to females with black hair, which had not been seen during training. The GAN models are less successful at modeling this transformation on out-of-sample data. In the parts of the image that should stay the same (everything but the hair color), they do not always generate a recreation of the input. In the hair color, only sometimes is the color changed. The regular GAN model especially has copious artifacts that are a result of the difficulty in training these models. This provides further evidence of the benefits of avoiding these complications when possible, for example by using the stable training of an autoencoder and editing it as we do in neuron editing.

We quantify the success of neuron editing by using the common metric of Frechet Inception Distance (FID) that measures how well the generated distribution matches the distribution targeted for extrapolation. These scores are reported in Table 1, where we see neuron editing achieve the best result on an average of three runs. Notably, due to the autoencoder's more stable training, the standard deviation across multiple runs is also lower than the GAN-based methods.

In Figure 2b, we motivate why we need to perform the $NeuronEdit$ transformation on the internal layer of a neural network, as opposed to applying it on some other latent space like PCA. Only in the neuron space has this complex and abstract transformation of changing the hair color (and only the hair color) been decomposed into a relatively simple and piecewise linear shift.

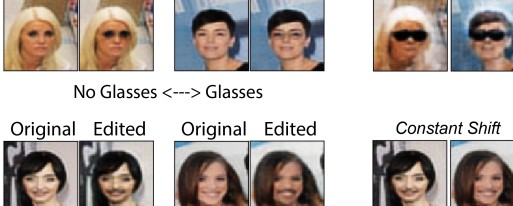

Figure 3: Additional CelebA transformations.

Beyond hair color transformation, neuron editing is able to learn general transformations on CelebA males and apply them to females. In Figure 3, we learn to transform between having/not having the mustache attribute and having/not having the glasses attribute. The latter transformation on glasses demonstrates the importance of learning a non-constant transformation. The glasses attribute is bimodal, with both examples of sunglasses and reading glasses in the dataset. With neuron editing, we are able to learn to map to each of these different parts of the latent space, as opposed to the constant shift which adds dark sunglasses to the entire distribution.

### 3.2 Batch correction by out-of-sample extension from spike-in samples

We next demonstrate another application of neuron editing's ability to learn to transform a distribution based on a separate source/target pair: biological batch correction. Many biological experiments involve using an instrument to measure different populations of cells and then characterizing the features that distinguish between them. However, these complex instruments can be difficult to calibrate and use consistently, and thus can introduce technical artifacts into the data they are used to measure. In fact, we can even measure the same population of cells twice and get two very different datasets back. When we measure *different* populations, these technical artifacts (batch effects) get confounded with the true differences between the populations. Batch effects are a ubiquitous problem in biological experimental data can lead to incorrect conclusions in downstream analysis. Addressing batch effects is a goal of many new models (Finck et al., 2013; Tung et al., 2017; Butler & Satija, 2017; Haghverdi et al., 2018), including some deep learning methods (Shaham et al., 2017; Amodio et al., 2018).

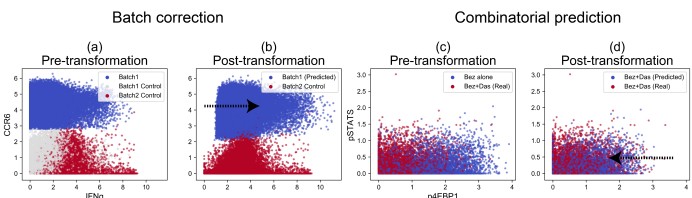

Figure 4: Neuron editing corrects the variation in IFNg while preserving the variation in CCR6 and correctly predicting the effect of combining two drugs.

| Neuron Editing | GAN | CycleGAN | ResnetGAN | RegAE | Constant Shift | CCA | MNN | ComBat | Limma |
|---|---|---|---|---|---|---|---|---|---|
| **0.96275** | 0.5108 | 0.4310 | 0.6268 | -0.0508 | 0.88205 | 0.6034 | 0.5339 | 0.5569 | 0.5431 |

Table 2: Correlation between observed change in spike-ins and applied change to samples. Neuron editing most accurately applies just the transformation observed as batch effect and not true biological variation.

One method for grappling with this issue is to repeatedly measure an unvarying control (called a spike-in) set of cells with each population of interest (called a sample) (Bacher & Kendziorski, 2016). Because we know any observed differences in the spike-in are technical artifacts, we can model and then remove this artifact in the population of interest. In our previous terminology, the two spike-in distributions are our known source/target pair while the actual population of interest is our second source that lacks a known target.

Existing methods of batch correction based on spike-ins work directly in the data space, operate independently on each dimension, and only do crude matching of distribution statistics. The most common approach is to simply subtract the difference in means between the spike-ins from the sample. We believe this is natural opportunity for deep learning, where the same concept can be extended to an abstract feature space, composed of combinations of features, and a more powerful transformation. Moreover, we expect neuron editing to shine as the spike-ins likely differ drastically from the sample.

The dataset we investigate in this section comes from a mass cytometry (Bandura et al., 2009) experiment which measures the amount of particular proteins in each cell in two different individuals infected with dengue virus (Amodio et al., 2018). We note that these data are in a drastically different format from the images of the previous experiment, as they are in tabular form with cell $i$ being row $i$ and the amount of protein $j$ in column $j$. We believe a key strength to neuron editing is its general applicability to a wide range of data types and modalities. In this particular experiment, there are four datasets, each consisting of measurements of 35 proteins: the two spike-ins we refer to as Control1 and Control2 are shape $18919 \times 35$ and $22802 \times 35$, respectively, while the two populations we actually want to study, called Sample1 and Sample2, are shape $94556 \times 35$ and $55594 \times 35$. To better grasp the problem of batch effects, we visualize a biaxial plot with two of the proteins where there is a batch effect in one dimension and a true underlying biological difference in the other dimension (Figure 4). By using the controls, we seek to correct the artificially low readings of the protein IFNg in Sample1 (along the x-axis) without removing the biologically accurate readings of higher amounts of protein CCR6 (along the y-axis).

We would like our model to identify this source of variation and compensate for the lower values of IFNg without losing other true biological variation in Sample1. For example, Sample1 also has higher values of the protein CCR6, and as the controls show, this is a true biological difference, not a batch effect (the y-axis in Figure 4a).

We quantify the performance of the models at this goal by measuring the correlation between the change in median marker values observed in the spike-in with the change applied to the sample. If this correlation is high, we know the transformation applied to the samples only removes the variation where we have evidence, coming from the spike-ins, that it is a technical artifact. This data is presented in Table 2, where we compare to not only the deep generative models we have already introduced, but also dedicated batch correction methods commonly used by practitioners (Johnson et al., 2007; Haghverdi et al., 2018; Butler & Satija, 2017). We see that neuron editing outperforms

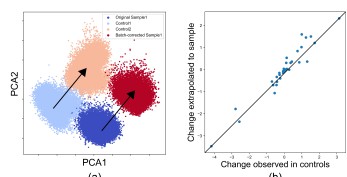

Figure 5: (a) The global shift in the two controls (light blue to red) is isolated and this variation is edited into the sample (dark blue to red), with all other variation preserved. (b) The median change in the sample in each dimension corresponds accurately with the evidence in each dimension in the controls.

| | Neuron Editing | GAN | CycleGAN | ResnetGAN | RegAE | Constant Shift |
|---|---|---|---|---|---|---|
| $r$ | **0.99661/0.97232** | 0.87014/0.58130 | 0.92687/0.85380 | 0.94529/0.93032 | 0.91965/0.96053 | 0.96680/0.96925 |

Table 3: Correlation between real and predicted means/variances on the combinatorial drug prediction data. The GANs generate data that is less accurate (means are off) and less diverse (variances are smaller) than the real data, while neuron editing best models the true distribution.

all of the alternatives at extrapolating from the spike-ins to the samples. This is unsurprising, as the GAN methods are only trained to produce data like Control2, and thus will not preserve much of the variation in the sample. The traditional batch correction methods make specific parametric distributional assumptions on the data that are not held in practice, and thus also perform poorly. The regularized autoencoder, since the transformation is performed during training rather than after training like neuron editing, just reproduces its input unchanged.

In Figure 5a, a PCA embedding of the data space is visualized for Control1 (light blue), Control2 (light red), Sample1 (dark blue), and post-transformation Sample1 (dark red). The transformation from Control1 to Control2 mirrors the transformation applied to Sample1. Notably, the other variation (intra-sample variation) is preserved. In Figure 5b, we see that for every dimension, the variation between the controls corresponds accurately to the variation introduced by neuron editing into the sample. These global assessments across the full data space offer additional corroboration that the transformations produced by neuron editing reasonably reflect the transformation as evidenced by the controls.

### 3.3 COMBINATORIAL DRUG TREATMENT PREDICTION ON SINGLE-CELL DATA

Finally, we consider biological data from a combinatorial drug experiment on cells from patients with acute lymphoblastic leukemia (Anchang et al., 2018). The dataset we analyze consists of cells under four treatments: no treatment (basal), BEZ-235 (Bez), Dasatinib (Das), and both Bez and Das (Bez+Das). These measurements also come from mass cytometry, this time on 41 dimensions, with the four datasets consisting of 19925, 20078, 19843, and 19764 observations, respectively. In this setting, we define the source to be the basal cells, the target to be the Das cells, and then extrapolate to the Bez cells. We hold out the true Bez+Das data and attempt to predict the effects of applying Das to cells that have already been treated with Bez.

Predicting the effects of drug combinations is an application which is typically approached through regression, fitting coefficients to an interaction term in a multiple linear regression model. This limitation of only fitting linear relationships and treating each protein independently, greatly restricts the model in a biological contexts where we know nonlinearity and protein regulatory networks exist and play a large role in cellular function. Using neuron editing in this context facilitates learning a much richer transformation than previous, non-deep learning methods.

We quantitatively evaluate whether neuron editing produces a meaningful transformation in Table 3, where we calculate the correlation between the real and generated means and variances of each dimension. Neuron editing more accurately predicts the principle direction and magnitude of transformation across all dimensions than any other model. Furthermore, neuron editing better preserves the variation in the real data. The GANs have trouble modeling the diversity in the data, as manifested by their generated data having significantly less variance than really exists.

We see an example of the learned transformation by looking at a characteristic effect of applying Das: a decrease in p4EBP1 (seen on the x-axis of Figure 4c). No change in another dimension, pSTATS, is associated with the treatment (the y-axis of Figure 4c). Neuron editing accurately models this change in p4EBP1, without introducing any change in pSTATS or losing variation within the extrapolation dataset (Figure 4d).

We note that since much of the variation in the target distribution already exists in the source distribution and the shift is a relatively small one, we might expect the ResnetGAN to be able to easily mimic the target. However, despite the residual connections, it still suffers from the same problems as the other models using the generating approach: namely, the GAN objective encourages all output to be like the target it trained on. This leaves it unable to produce the correct distribution if it differs from the target of the learned transformation, as we see in this case.

## 4 DISCUSSION

In this work, we have only consider learning from a single pair of distributions and applying it to another single distribution. We consider it an interesting direction for future work to extend this to multiple distributions, either for learning from and application to. Additional future work along these lines could include training parallel encoders with the same decoder, or training to generate conditionally.

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
