# OpenReview forum: "Beyond GANs: Transforming without a Target Distribution"
_ICLR.cc/2020/Conference — Reject_

### Official Review · AnonReviewer3 · 2019-10-23
**Official Blind Review #3**

**Rating:** 6

**Review:**

This paper tackles the problem of mismatched training/test data by directly modifying the latent representation learned with an auto-encoder. The proposed method employs a piece-wise linear function to transform the representation of samples from a source distribution to that of samples from the corresponding target distribution. The paper argues that the same transformation can be applied to samples from a distribution that are different from the source/target distributions. The method is empirically justified by three different tasks in computer vision and biology.

In my opinion, the motivation of the paper is clear and the writing is easy to follow, but one potential limitation is the lack of comparison with recent related work, e.g.,

Generate To Adapt: Aligning Domains using Generative Adversarial Networks
Swami Sankaranarayanan, Yogesh Balaji, Carlos D. Castillo, Rama Chellappa
https://arxiv.org/abs/1704.01705

Deep Transfer Learning with Joint Adaptation Networks
Mingsheng Long, Han Zhu, Jianmin Wang, Michael I. Jordan
https://arxiv.org/abs/1605.06636

**Experience Assessment:**

I do not know much about this area.

**Review Assessment: Checking Correctness Of Derivations And Theory:**

I carefully checked the derivations and theory.

**Review Assessment: Checking Correctness Of Experiments:**

I assessed the sensibility of the experiments.

**Review Assessment: Thoroughness In Paper Reading:**

I read the paper at least twice and used my best judgement in assessing the paper.

---

> ### Author Response · Authors · 2019-11-13
> **Response to Reviewer#3**
>
> Reviewer#3, thank you for your recommendation of accepting our paper, and thank you for the time and energy spent carefully reading it! We are glad that you found our writing clear and a valuable contribution to the literature.
>
> We also appreciate being pointed to these relevant works. While our technique is different, there are definitely important connections to previous domain adaptation work, and we have used these to better place where neuron editing falls in the landscape of learning transformations that can extrapolate [1, 2]. Domain adaptation seeks to learn a network on some labeled data that can be applied to some unlabeled data. Almost exclusively, these works consider the task to be *classification*, and labels are the *class labels*, which are known on one dataset and not the other. Instead, neuron editing considers the task *transformation*, and labels are the *target distribution*, which is known for one dataset and not the other. Thus, neuron editing is a transformation-learning extension of the existing classification-learning domain adaptation work.
>
> We had not thought of it in such terms before, but greatly appreciate your recommendation. Our manuscript is better off for the inclusion, so we thank you again!
>
> [1] Tzeng, Eric, et al. "Adversarial discriminative domain adaptation." Proceedings of the IEEE Conference on Computer Vision and Pattern Recognition. 2017.
> [2] Ghifary, Muhammad, et al. "Deep reconstruction-classification networks for unsupervised domain adaptation." European Conference on Computer Vision. Springer, Cham, 2016.

---

### Official Review · AnonReviewer2 · 2019-10-24
**Official Blind Review #2**

**Rating:** 6

**Review:**

In this paper, the authors present a method for learning a transformation between two distributions and applying it to an out-of-sample extrapolating distribution. The method relies on an autoencoder, instead of a GAN, to which neuron-editing is applied. A transformation of one of the inner layer is learned using the source and target distributions is estimated and then applied to the extrapolating distribution. This leads to better reconstructions than GAN approaches as judged by some visualizations and results. The method is also applied to biological datasets and some improvement is shown (accuracy increases on specific prediction tasks).

The idea of applying neuron-editing to an autoencoder is pretty interesting. It's a simple manipulation that makes a lot of sense and judging by the image transformation examples works well. This method also numerically greatly outperforms others on the CelebA extrapolation task so the extrapolation is believable, even though more examples in the appendix would be good. The motivation of applying the method to medical data to correct for instrument variability is also very interesting.

However I felt that I could not fully see the benefit of the application for medical data because the area, task, datasets etc are not well introduced. I think the datasets should be explained better, and examples of the images should be given. I understand the gist of Figure 4 but it's not well explained and I do not see why these dimensions were picked. I think there is more work to do there. I think a more careful introduction to the field, with explanations of the data, why deep methods are applicable there, what people have tried etc are necessary.

Also all of the tables in the paper with classification tasks should have sections in the appendix to explain everything about those tasks. In general, it seems the main weakness of the paper is in exposing the information/writing.

Smaller points:
In section 2, source, target and extrapolation distributions are not introduced properly. In equation 1 and the text around it, it's hard to tell that each dimension is edited independently without a couple of reads.
" biologicl batch correction" page 6
I think the second paragraph in page 4 ("To apply the learned...") is missing a sentence in the middle about the actual editing step.

Perhaps more can be said about multiple extrapolation datasets (measurements from different dates instead of only two), if possible/available.

**Experience Assessment:**

I have read many papers in this area.

**Review Assessment: Checking Correctness Of Derivations And Theory:**

I assessed the sensibility of the derivations and theory.

**Review Assessment: Checking Correctness Of Experiments:**

I assessed the sensibility of the experiments.

**Review Assessment: Thoroughness In Paper Reading:**

I read the paper at least twice and used my best judgement in assessing the paper.

---

> ### Author Response · Authors · 2019-11-13
> **Response to Reviewer#2**
>
> Reviewer#2, we thank you for the recommendation to accept our paper and are grateful for the time and effort in carefully reading it! We are glad you are able to appreciate the contribution it would make to the discussion regarding generative modeling. We thank you for your one major critique regarding the clarity of the biological applications. We have made this explanation much more comprehensive in our revision and will gladly continue to improve it as we receive recommendations. We have explained more about the particular mass cytometry datasets, the task of batch correction  itself, the most pressing computational problems of existing approaches, and why we believe deep learning is an appropriate tool. Specifically, we have heavily edited the entirety of Section 1 and the first several paragraphs in both Sections 3.2 and Section 3.3. As these important applications are a key use of our technique in our actual work, we want to make sure they are motivated and described as clearly as possible!
>
> To the small points:
> — Specifically, to the figure you mention, Figure 4, this visualization was chosen for a few main reasons. First, biaxial plots are a commonly used way of visualizing cellular (cytometry) data of this kind [1]. Second, viewing the raw data values is important for verifying that neuron editing’s batch-correction transformations are *biologically sensible*. For example, when defending a biological conclusion based on the batch-corrected data, this visualization would be necessary to justify why interferon-gamma values in the sample were increased (because the technical replicate spike-ins showed that batch measured artificially low interferon-gamma).
> — Thank you for the notes about the minor edits here and there.
> — The idea of extrapolating to more than just one pair is a very interesting one, and one that we think would be promising future work, that’s a great insight, which we have included in our final discussion.
>
> Again, thank you for your review and the help improving our manuscript!
>
> [1] Korin, Ben, Tania Dubovik, and Asya Rolls. "Mass cytometry analysis of immune cells in the brain." Nature protocols 13.2 (2018): 377.

---

### Official Review · AnonReviewer1 · 2019-10-25
**Official Blind Review #1**

**Rating:** 3

**Review:**

Overview: This paper presents a new generative modeling approach to transform between data distributions via a technique the authors dub “neuron editing”. Their approach for “neuron editing” can be used to learn how neurons in a DNN encode particular transformations in the latent space, with the hope of using it to be able to generate data that are out-of-sample and/or do not lie on the same manifold as the data used to train the generative model. Experiments were performed on the CelebA dataset. The authors further demonstrate the usefulness of their proposed idea by using it to remove noise/batch effects from data and also predicting synergy between drugs by using their idea to model the effect of drug treatments.

While the idea behind this paper is interesting and their method does seem to provide improvements over other generative models, the paper is a bit difficult to follow and the applications to biology are not explained well enough to understand the implications of their results. The fact that the authors did not compare to StyleGANs also seems a bit suspicious because these models are sota for the types of problems that this paper addresses. Additionally, the authors should consider rewriting the motivation for this problem to be more general. This is a convincing new method to transform distributions in any setting, but the introduction would lead one to believe its applicability is mostly to computational biology.

Detailed comments:
- The writing and motivation needs to be reworked in order to ensure that the introduction, results, and abstract match in their tone and content. The abstract uses a face recognition/generation application to motivate the work, the introduction focuses solely on computational biology, the results switch between biology and face recognition.

- Section 2: Three desirable properties for a transformation are listed, but what makes these properties desirable?
Equation (1): Not obvious why the piecewise function definition is necessary, since the second equation seems to hold for j = 0 and j = 99 as well. This should either be corrected or simplified.

- “Mode collapse refers to the discriminator being unable to detect differences in variability between real and fake examples.” Not true. Mode collapse is when a large region of the model’s input space maps onto the small region around a single (often bad) sample.

- Section 3: It would be helpful to see gradual interpolation from the original latent representation of a sample along the direction the neuron edit will be performed, and continued beyond the proposed final latent activations, to demonstrate how this process actually affects the model’s output.

- Doesn’t really seem like the authors tried very hard with the other methods (especially the vanilla GAN) and the omission of StyleGAN, the current state-of-the-art in this kind of transformation learning, is conspicuously omitted. It is highly likely that a well-trained StyleGAN would do better than the other GAN/AE techniques compared against.

- A PCA-based transformation is applied directly to the image data, but it would be interesting (and perhaps more informative) to perform an alignment of the PCs of activations in the latent space.

- Section 4: Again, the motivation distinctly focuses on computational biology when one could easily imagine this approach being applicable to a variety of problems.

- Comparisons with StyleGANs would’ve been appreciated, especially given the fact that they’re now considered state-of-the-art when it comes to modifying the latent space and creating out of sample images as proposed in this work.

- The idea is interesting, though one wonders if there isn’t any other work where the neurons have been “edited” to accommodate different transformations, given that the idea is itself rather intuitive. A more thorough literature review in that regard would be helpful.

- The CelebA experiments help lay establish an intuitive understanding of the proposed technique and were helpful. However, the ideas are a little disconnected with the biological applications of the technique. Better motivation/bridging of the two sets of experiments would be nice.

- A more detailed explanation and analysis of the combinatorial drug application would’ve been helpful to understand the results.


**Experience Assessment:**

I have read many papers in this area.

**Review Assessment: Checking Correctness Of Derivations And Theory:**

N/A

**Review Assessment: Checking Correctness Of Experiments:**

I assessed the sensibility of the experiments.

**Review Assessment: Thoroughness In Paper Reading:**

I read the paper at least twice and used my best judgement in assessing the paper.

---

> ### Author Response · Authors · 2019-11-13
> **Response to Reviewer#1**
>
> Reviewer#1, thank you for such a detailed review! We hope that you will reconsider your rating in light of our comments and revision, and join the other two reviewers in recommending our paper for acceptance.
>
> Your point that the introduction was far more narrowly tailored (to the biological application) than the content of the paper itself was extremely helpful. We have rewritten the introduction to speak to the method’s general applicability, as opposed to motivating it only for the context of computational biology. Specifically, we have heavily edited the entirety of Section 1 and the first several paragraphs in both Sections 3.2 and Section 3.3, hopefully making these aspects far more clear than they were in their first version. We think this significantly improves the cohesion and consistency of the presentation. However, it is worth mentioning that the biological application is very compelling and solves long standing problems in both batch correction and predicting combinatorial drug effects.
>
> We are not sure we agree about the applicability of the recently published StyleGAN to our problem, however. While its results are definitely impressive, if we understand correctly, StyleGAN is used for generation based on a code sampled from a noise distribution Z (like a standard Gaussian). The results are individual high quality images from scratch, but this is a fundamentally different task than ours. Neuron editing’s task is to take as input a data point from one dataset (and no noise), and generate a data point from another dataset (referred to as domain mapping, image-to-image translation, manifold alignment, or any of a number of other names). This is different from generating just any out-of-sample point, but we would need to generate one from a given input point. While the paper does mention a sort of “style mixing”, this requires having a known source point in the other domain, for one thing, and appears to be a non-standard use of the model anyway. We believe that the gold-standard baseline for the task of domain translation is the CycleGAN, which we compare to in our experiments.
>
> In response to the small points you helpfully raised:
> — We think our definition of mode collapse (tied to a notion of diversity in the real dataset that is unmodeled by the generator) is still a correct description, albeit from a different perspective than it is usually thought of. Mode collapse doesn’t just refer to a large part of the latent space mapping to a small region around a single point (and it especially doesn’t require the generated point to be inaccurate). This definition is biased by our use of datasets which are standard collections of images: consider for example if the target dataset is *in fact* just a small region around a single point. Modeling this effectively obviously wouldn’t be mode collapse, even though the whole latent space would map to a small region. We proffer that mode collapse requires a notion of *diversity in the dataset* that isn’t being modeled. But this is a minor quibble where we can reasonably disagree and would gladly reword the offending line if desired.
> — The diversity of data types we experiment on are a strength of the technique, and appreciate the suggestions on how to better bridge the discussion/motivation of these sections.
> — Regarding previous work, we briefly mentioned the most related works that assume the shift in latent space is a single constant vector across the whole distribution, and that use highly specialized pre-trained models on labeled data to extract features in a supervised manner rather than our unsupervised autoencoder. We could expand on these sections as necessary.
>
> Overall, we greatly appreciate your feedback and hope you believe our responses and revisions have addressed your concerns sufficiently to upgrade your rating. Thank you for your role in improving our manuscript!

---

### Decision · Program_Chairs · 2019-12-19

**Decision:**

Reject

**Comment:**

This paper presents a new generative modeling approach to transform between data domains via a neuron editing technique. The authors address the scenario of source to target domain translation that can be applied to a new source domain. While the reviewers acknowledged that the idea of neuron editing is interesting, they have raised several concerns that were viewed by AC as critical issues: (1) given the progress that have been made in the field, an empirical comparison with SOTA GANs models is required to assess the benefits/competitiveness of the proposed approach -- see R1’s comments, also [StarGAN by Choi et al, CVPR 2018], (2) the literature review is incomplete and requires a major revision -- see R1’s and R3’s suggestions, also [CYCADA by Hoffman et al, ICML 2018], (3) presentation clarity -- see R1’s and R2’s comments. AC suggests, in its current state the manuscript is not ready for a publication. We hope the detailed reviews are useful for improving and revising the paper.